# Comparative Clinical Study of Suprathel^®^ and Jelonet^®^ Wound Dressings in Burn Wound Healing after Enzymatic Debridement

**DOI:** 10.3390/biomedicines11102593

**Published:** 2023-09-22

**Authors:** Wolfram Heitzmann, Mitja Mossing, Paul Christian Fuchs, Jan Akkan, Harun Seyhan, Gerrit Grieb, Christian Opländer, Jennifer Lynn Schiefer

**Affiliations:** 1Clinic of Plastic, Reconstructive, Hand and Burn Surgery, Cologne-Merheim Hospital, Witten/Herdecke University, 51109 Cologne, Germany; heitzmannw@kliniken-koeln.de (W.H.); akkanj@kliniken-koeln.de (J.A.);; 2Department of Plastic Surgery and Hand Surgery, Gemeinschaftskrankenhaus Havelhoehe, Teaching Hospital of the Charité Berlin, Kladower Damm 221, 14089 Berlin, Germany; gerritgrieb@gmx.de; 3Clinic of Plastic Surgery, University Hospital Aachen, Pauwelsstraße 30, 52074 Aachen, Germany; 4Institute for Research in Operative Medicine (IFOM), Cologne-Merheim Hospital, Witten/Herdecke University, Ostmerheimer Str. 200, 51109 Cologne, Germany

**Keywords:** Jelonet^®^, Suprathel^®^, deep dermal burns, enzymatic debridement, wound healing

## Abstract

Following the enzymatic debridement of deep dermal burns, the choice of wound dressing is crucial for providing an adequate environment and suitable conditions for rapid wound healing. As Suprathel^®^ and fatty gauze (Jelonet^®^) are the most commonly used dressings in burn centers, the aim of this study is to compare Suprathel^®^ and Jelonet^®^ in the treatment of deep dermal burns after enzymatic debridement with respect to wound healing, patient comfort, and pain. A total of 23 patients with deep dermal burns of the hand or foot (mean total body surface area of 4.31%) were included in this prospective, unicentric, open, comparative, and intra-individual clinical study. After enzymatic debridement, wounds were divided into two areas: one was treated with Suprathel^®^ and the other with Jelonet^®^. Suprathel^®^ remained on the wounds without dressing changes while Jelonet^®^ was regularly changed. Wound healing, infection, bleeding, exudation, time for dressing changes, and pain were documented (from days 2 to 48) after injury. Satisfactory results were obtained in 22 cases; only one patient had to undergo a second debridement followed by skin grafting. No significant difference in time to final wound healing could be observed (18–19 d). Patients reported significantly less pain during the dressing changes for Suprathel^®^ compared to Jelonet^®^. Furthermore, the wound areas treated with Suprathel^®^ showed significantly less exudation and bleeding. Wound infections rarely occurred in both groups. In conclusion, the authors found that both wound dressings could be used to achieve safe and rapid wound healing after the enzymatic debridement of deep dermal burns of the hands and feet. However, compared to Jelonet^®^, Suprathel^®^ showed superior results in terms of patient comfort and pain reduction.

## 1. Introduction

Burn injuries are the fourth most common type of injury found worldwide [1], with 41.5% of burns affecting the hand, either as part of a more extensive burn or as an isolated injury. The most common complications following a hand burn are scar disturbances and scar contractures [2]. Often, an impairment in daily activities, quality of life, and work productivity can still be seen in burn patients 5–7 years after the trauma [3]. To prevent these complications, the rapid removal of burn eschar is the initial step in the treatment of deep burn wounds [4]. In doing so, vital tissue is preserved, as saving the stem cells within the deep dermal layers especially has been shown to be a key aspect of achieving satisfying long-term results [5,6].

Even though the ideal timing for the first debridement has been continuously discussed, early tangential excision followed by skin grafting has been the standard therapy for deep dermal and full-thickness burn wounds for decades [7]. However, this debridement method had disadvantages such as causing huge trauma and excessive bleeding, as well as being nonselective and removing vital tissue, leading to poor aesthetic and functional outcomes [8].

With NexoBrid**^®^** (MediWound, Rüsselsheim, Germany), a debridement agent based on a mixture of proteolytic enzymes that was introduced in the European market in 2012, the enzymatic debridement method soon superseded tangential excision as the gold standard in burn centers for deep dermal hand, foot, or facial burns. Various studies confirmed that this new technique was much more selective than surgical tangential debridement [9,10,11,12]. Thus, the long-term scar quality in partial-thickness-to-deep-dermal burn wounds was found to improve compared to traditional surgical tangential debridement [13]. In addition, the need for secondary reconstructive surgery due to burn contractures was shown to be decreased in patients treated with NexoBrid**^®^** compared to patients who received traditional surgical treatment [14]. However, it has been shown that not only the method of debridement but also the choice of primary wound dressing after debridement is of utmost importance, as it can influence the time to wound healing, thereby affecting scarring and aesthetic and functional outcomes [15]. Frequent multi-professional expert panel meetings of European plastic surgeons and burn care specialists using NexoBrid**^®^** revealed the fact that most burn centers either used Suprathel**^®^** or Jelonet**^®^** to cover the wounds after enzymatic debridement [6,16].

Suprathel**^®^** (PolyMedics Innovations GmbH, Denkendorf, Germany), a temporary skin substitute, is a thin and flexible hydrolytic membrane that consists mainly of polylactid, trimethylencarbonat, and ε-Caprolacton. Its application to partial-thickness burn wounds showed a good impact on wound healing and pain reduction [17]. Suprathel**^®^** was also shown to be an appropriate dressing for wound treatment after enzymatic debridement. Complete healing (less than 5% rest defect) was achieved on average by day 28 [5]. Due to the high price of Suprathel**^®^**, the search for cost-effective alternatives with comparable properties is still ongoing [18]. Simply using Jelonet**^®^**, a leno-weave fabric of cotton impregnated with white soft paraffin to cover the wounds after NexoBrid**^®^** treatment, has also become standard in many German burn centers. However, to date, no studies including a direct intra-individual comparison have been conducted. The aim of this study, therefore, is to compare, for the first time, the expensive Suprathel**^®^** to the cost-effective Jelonet**^®^** in the treatment of deep dermal burns after enzymatic debridement to see whether Suprathel**^®^** has a beneficial effect as far as wound healing, patient comfort, and pain are concerned.

## 2. Materials and Methods

The present study prospectively evaluates the healing of partial-thickness-to-deep-dermal burn wounds after enzymatic debridement followed by immediate treatment with Suprathel**^®^** and Jelonet**^®^** in a clinical setting.

### 2.1. Patients

All patients aged 18 to 85 years, who sustained partial-thickness-to-deep-dermal flame, scald, or contact burns of their hands or feet with more than 0.3% of their total body surface area (TBSA) affected, were enrolled after consenting to participate in this prospective, unicentric, open, comparative, and intra-individual clinical study. The exclusion criteria were a lack of consent and compliance in the follow-up examinations, burns caused by chemical substances or electricity, full-thickness burns, localization of the burned area to the face, an ABSI score of 10 or more, and pregnancy or nursing. A total of 23 patients meeting the eligibility criteria were enrolled between August 2020 and December 2022. Demographic data and data on the medications administered by the treating emergency physicians were collected and documented.

Prior to enrolling patients, the study was reviewed and approved by the Ethical Review Committee of the University of Witten Herdecke, Germany (protocol no. 38/2020) and all patients provided written informed consent. The study protocol conformed to the ethical guidelines of the 1975 Declaration of Helsinki.

### 2.2. Treatment

On the day of admission, all wounds were cleaned manually with cotton gauze using a Prontosan**^®^** (B. Braun Melsungen AG, Melsungen, Germany) wound irrigation solution. The TBSA and burn depth were then estimated by a burn surgeon. When the burn depth was evaluated as partial-thickness-to-deep-dermal, wound debridement was performed enzymatically. In the current study, a partially purified proteolytic protein mixture derived from pineapple plant stems was used (NexoBrid**^®^**, Mediwound, Yavne, Israel). Thus, the burn eschar was selectively dissolved while the vital and healthy tissue was preserved [19]. Enzymatic debridement was performed following a three-day treatment procedure comprising a prolonged pre- and post-soaking time overnight. Debridement took place under local or regional anesthesia and analgosedation following overnight soaking on the second day after burn injury [13]. Afterward, further wound treatment had to be performed according to the wound depth. Depth was assessed after the removal of the occlusive dressing. In the case of full-thickness wounds, the wound has to be covered by a skin transplant [5,13]. If the attending physician assessed the depth as a partial-thickness-to-deep-dermal burn and the patients agreed to take part in the study, the wound would be divided into two areas: one would be treated with Suprathel**^®^** and the other with Jelonet**^®^** (Smith & Nephew, Watford, UK) simultaneously. After application, the wounds were covered with a Prontosan**^®^**-impregnated cotton gauze and an external dressing. The wound regime was analogous for both dressings in accordance with our standard of care, except for the fact that Jelonet**^®^** was changed every two days, whereas Suprathel**^®^** remained until complete wound healing and detached itself independently after complete re-epithelialization.

### 2.3. Evaluation of Wound Parameters

The primary outcome measures investigated in this trial were infection, bleeding, exudation, and pain. The necessary time for dressing changes was also documented. The occurrence of an infection, exudation, or bleeding was analyzed on days 1, 2, 4, 8, 12, 16, 24, and 48 by visual inspection. The pain was also analyzed on days 1, 2, 4, 8, 16, 24, and 48 by using a verbal rating scale (VRS) for pain (0 = no pain to 10 = the most extreme pain), as reported by the patient [20]. The secondary outcome investigated in this study was the time from wound treatment until wound healing, which was defined as less than a 5% residual wound defect.

### 2.4. Statistical Analysis

Microsoft Excel (2017, Microsoft, Redmond, WA, USA) was used for data analysis and chart creation. After a thorough review of all the data, SPSS (Version 21, IBM, Armonk, NY, USA) was used for the final statistical analysis. Statistical significance was accepted at *p* ≤ 0.05.

With 20 pairs of data, a difference of two-thirds of SD could be detected (α < 0.05) with sufficient power (80%). As 23 patients happened to match the eligibility criteria between August 2020 and December 2022, they were taken in, although the study was originally designed for 20 patients. Statistically significant differences between the subgroups were identified with the pared *t*-test, the Wilcoxon test, and an AUC (area under the curve) analysis.

## 3. Results

Among the 23 patients who completed the trial, the partial-thickness-to-deep-dermal burns were mainly caused by scald (43.5%), followed by flame (34.8%). The average TBSA was 4.31%, with a minimum of 0.3% and a maximum of 23%. The proportions of males, females, and transgender people were 78.3%, 17.4%, and 4.3%, respectively. The mean age was 39.17 years (Table 1).

### 3.1. Wound Healing

Wound healing was achieved in 22 cases; only one patient had to undergo a second debridement followed by skin grafting after the initial use of Suprathel*^®^*. Once placed on the wounds, Suprathel*^®^* was gradually cut back as re-epithelialization progressed until the dressings were completely detached. So, the primary dressings did not need to be changed during the study period. In contrast, the Jelonet*^®^* dressings were changed as planned every two days. However, a significant difference in the application time between the two dressings was not observed. Wound closure was documented with a mean of 18.44 ± 4.78 days for wounds treated with Suprathel*^®^* and 18.81 ± 5.74 days for wounds treated with Jelonet*^®^* (*p* = 0.58); thus, there was no significant difference in time to the final wound healing (Figure 1).

### 3.2. Pain, Bleeding, Exudation, and Time Needed for Dressing Changes

Patients rated their pain level using the VRS (0 = no pain at all to 10 = extreme pain). Wound-related pain scores were low on day 1 (mean of 1.79 for both wound dressings), increased on day two, as the first secondary dressing change in the treatment with Suprathel*^®^* and the first complete dressing change with the Jelonet*^®^* treatment took place, and then decreased during the healing process (Figure 2A). Patients reported significantly less pain after the application of Suprathel*^®^* compared to the use of Jelonet*^®^* on day 2 (*p* < 0.001) and day 4 (*p* < 0.01).

Wounds, in general, showed significantly less exudation after applying Suprathel*^®^* than after applying Jelonet^®^ (*p* < 0.05). The most exudation in both groups occurred during the first secondary wound dressing change on day 2 (22 of 23 patients showed exudation in the area treated with Jelonet^®^ and 19 of 23 patients showed exudation in the area treated with Suprathel^®^). Exudation decreased during the healing process (Figure 2B).

After applying Suprathel*^®^* to the wound, no bleeding occurred during the whole healing process, whereas bleeding was most often seen in wound areas during the Jelonet*^®^* changes on day 2 and day 4 (Figure 2C). Therefore, the bleeding rates were significantly lower in the Suprathel^®^ areas (*p* < 0.001). Due to the controlled study setting there was no significant difference in the frequency and time needed for dressing changes between the wounds covered with Suprathel*^®^* and those covered with Jelonet*^®^*.

## 4. Discussion

Burn wounds are a very special entity of wounds, as there are many factors that have to be addressed such as burn depth, TBSA, or the type of burn (scald, contact burn, or flame). Thus, they need particular attention and individual treatment plans. That is why much effort has been put into the development of new dressings to optimize burn wound treatment in recent decades. One of the most promising products seems to be the hydrolytic skin substitute made of a copolymer based on dl-lactic acid, Suprathel**^®^** [17]. To find proof of the advantage of Suprathel**^®^**, many studies comparing its effect on wound healing and scarring to other dressings have been conducted. However, most of these studies targeted the treatment of superficial-to-partial-thickness burn wounds. Thus, in 2022 our group compared the scar quality after treatment of superficial burns with Dressilk**^®^** and Suprathel**^®^**, finding that both wound dressings led to esthetically satisfying scar recovery without significant differences from normal uninjured skin after 12 months [21]. Another interesting wound dressing frequently used in burn wounds is Mepilex Ag. In 2018, Mepilex Ag was compared to Suprathel**^®^** in a prospective, randomized, and controlled trial by Hundeshagen et al., and it was found that both dressings had comparable healing periods as far as partial-thickness burns were concerned [22]. In comparing Suprathel**^®^** to Epicite**^®^**, we recently showed that burn wounds treated with these two dressings had comparable wound healing periods [18]. Rashaan et al. also showed that Suprathel**^®^** could safely be used in toddlers and children [23].

Taking all these findings into consideration, one can say that using Suprathel**^®^** is a safe and effective method to treat superficial-to-partial-thickness burn wounds but is much more expensive than using the other more cost-effective alternatives. Furthermore, Suprathel**^®^** has no advantages compared to the alternatives when it comes to superficial-to-partial-thickness burn wounds as far as wound healing time and scarring are concerned. 

Thus, the new question to be answered is whether Suprathel**^®^** can also be used in the treatment of more deeply burned wounds, where other dressings fail. As soon as the burn depth reaches the deep dermal layer, a sort of eschar removal is needed. The gold standard is tangential excision followed by split-skin grafting. Keck et al. showed that Suprathel**^®^** was an alternative to split-skin grafting in deep partial-thickness defects, showing longer healing periods in comparison to skin grafts but comparable results concerning early scar formation [24]. However, as far as hand and face burns are concerned, enzymatic debridement has replaced tangential excision as the gold standard nowadays [4]. At the Cologne burn center, we were able to develop a post-debridement wound treatment algorithm by using Suprathel**^®^** on deeply burned hands, showing promising results regarding handling as well as the duration of the treatment, efficiency and selectivity of debridement, healing potential, and early rehabilitation of the patients [5]. Dadras et al. also confirmed that Suprathel**^®^** could safely be used after enzymatic debridement. However, they did emphasize that this combined treatment required experience [25].

To the best of our knowledge, in German burn centers, enzymatically debrided deep dermal burn wounds are usually covered with either Suprathel**^®^** or fatty gauze (Jelonet**^®^**). However, studies including a direct comparison of these two dressings after enzymatic debridement have not yet been conducted. Therefore, the aim of this study was to compare Suprathel**^®^** to Jelonet**^®^** in the treatment of deep dermal burns after enzymatic debridement for the first time.

Deep partial-thickness burns reach into the deep dermis, damaging hair follicles and glandular tissue. If infection is prevented and the wounds are allowed to heal spontaneously without performing a skin graft, healing occurs in three to eight weeks with scarring [26,27]. In our study, wound healing was achieved in a mean of 19.58 ± 5.98 days for wounds treated with Suprathel**^®^** and 18.54 ± 5.75 days for wounds treated with Jelonet**^®^**, which indicated that both materials did not only secure but also accelerated wound healing. Another aspect that most likely plays an additional role in the short wound healing time is the fact that with an average TBSA of 4.31%, the burned areas are rather small. Here, no significant difference in wound healing time can be observed. Our findings are in accordance with the results of previous studies comparing Suprathel**^®^** to the other dressings mentioned above [18,21]. The fact that Suprathel**^®^** does not require a primary dressing change after application, unlike Jelonet**^®^** which is changed every two days, does not appear to have an impact on wound healing time.

In contrast to superficial partial-thickness burns, which are very painful from the beginning, deep partial-thickness burns present themselves as relatively painless at first. This is because nerve endings are also destroyed, leading to a decreased sensation [27]. In our study, patients rated their pain level by using the VRS (0 = no pain at all to 10 = extreme pain). Wound-related pain scores were low on day 1 (mean of 1.79 for both wound dressings) as expected. The first secondary dressing change at the Suprathel**^®^** area and, at the same time, the first primary dressing change of Jelonet**^®^**, took place on day 2. This procedure was shown to be the most painful event in the whole healing time (mean of 2.79 for Suprathel**^®^** and 3.71 for Jelonet**^®^**). However, patients reported significantly less pain after the application of Suprathel**^®^** compared to the use of Jelonet**^®^** (*p* < 0.001). These findings are in accordance with other studies that indicated Suprathel**^®^** has analgesic properties [22,28]. Another important aspect that explains the difference in the pain levels could be the fact that removing the first Jelonet**^®^** layer on day 2 might traumatize the recovering dermis, thereby leading to irritation and pain.

The main reason for morbidity and mortality after burn trauma is found to be burn wound sepsis. Infections by dangerous pathogens, such as Pseudomonas aeruginosa or Acinetobacter baumannii, delay patient recovery and can even lead to patient death [29]. This is partially due to an increased resistance to antibiotics which has been described in Pseudomonas aeruginosa and Acinetobacter baumanii in the context of burn wounds [30,31]. Wound infection, however, starts with colonization [32]. As a result of the reduced immune response, and because of the reduced blood supply and the nutrient-rich environment that can be found in burn wounds, a single bacterium can multiply into 10 million bacteria within 24 hours. If left untreated, this rapid colonization can lead to infection [33]. An increase in exudation has been shown to be a potential sign of acute wound infection [34]. In our study, burn wounds showed significantly lower exudation after applying Suprathel**^®^** than after applying Jelonet**^®^**. Regardless, exudation decreased during the healing process. Taking this into consideration, one possible explanation for these findings may be the antibacterial properties of the polylactic acid sheets as they seem to function as a barrier against bacterial penetration [35]. 

However, the infection rates in general were extremely low in this study; only two transient mild wound infections were found in each group. Thus, it may be assumed that Suprathel’s**^®^** beneficial effect on wound infections was equalized by regular Jelonet**^®^** dressing changes combined with wound disinfection in the areas that did not receive Suprathel**^®^**. In addition, it should also be taken into consideration that the low infection rates can partly be explained by the fact that the burned areas of the patients in this study were very small, with a mean TBSA of only 4.31%.

Furthermore, the regular changes of the primary dressing seemed to cause wound bed irritation, as significantly more bleeding was documented. After applying Suprathel**^®^** to the wound, no bleeding occurred during the whole healing process. These findings are in accordance with other studies where, for example, Suprathel**^®^** caused less bleeding than Mepilex**^®^** transfer did in the treatment of donor sites of split-thickness skin grafts [36].

One could, therefore, assume that the frequency and time for dressing changes were also reduced after the application of Suprathel**^®^**. However, there was no significant difference in either area. In general, the dressing changes after using Suprathel**^®^** are needed on days two and four, as well as once a week during the remaining healing time. In contrast, the dressing changes after using Jelonet**^®^** are needed more frequently, until full wound healing. Due to the controlled study setting, however, both areas were taken care of equally for the evaluation of bleeding and exudation so that no difference in dressing change frequency could be shown. As in most cases, only one hand was affected and the relatively small burn wounds on this hand were divided into two, so an adequate separation of the time periods needed for each dressing change was not always possible. In many cases, the Suprathel**^®^** area was covered with additional gauze only because the adjacent Jelonet^®^ area was still highly exudative, prolonging dressing changes for both areas. It can be assumed that in an inter-individual study setting, the frequency and time needed for dressing changes after applying Suprathel**^®^** can be reduced drastically compared to applying Jelonet**^®^**.

In a previous study from 2018, it was described that Suprathel**^®^** costs USD 0.56 per square centimeter [22]. If a hand measures approximately 100 square centimeters, covering it with Suprathel**^®^** would cost around USD 56. As Jelonet**^®^** was used regardless in this study, either as a secondary dressing on top of Suprathel**^®^** or as primary dressing directly on the wound, treating wounds with Suprathel**^®^** has higher material costs (USD 56). So generally, Jelonet^®^ is more cost-effective than Suprathel**^®^** as far as material costs are concerned. However, the manufacturer emphasizes that Suprathel**^®^** is more cost-effective than using fatty gauze [37] in terms of showing a trend toward the reduced demand for analgesics, reduced nursing time, and shorter hospital stays [17,28]. This fact might make up for the higher material costs in the long run.

### Limitations

As far as study limitations are concerned, the following aspects have to be taken into consideration. First, the study group was rather small and comprised only 23 patients. Multi-center studies with larger sample sizes are needed to validate our results. Furthermore, the burn depth was only assessed clinically. As our observed healing periods in general support the overall accuracy of our clinical judgment, it would be reasonable to involve objective assessment tools, such as laser Doppler, to further improve the diagnostic precision and group comparability in future studies. In addition to this, adjacent wound areas can still differ slightly in burn depth. Although the areas were chosen randomly, a physician’s tendency for applying Suprathel**^®^** to the wound area appearing more deeply burned could not be totally excluded. In addition, this study does not comprise long-term results and, therefore, no statement on the differences in scar quality can be made so far.

## 5. Conclusions

In conclusion, it can be said that both wound dressings are easily handled and can be used to achieve safe and rapid wound healing after the enzymatic debridement of deep dermal burns of the hands and feet. The less cost-effective Suprathel^®^, however, was shown to be superior as far as patient comfort and pain reduction were concerned.

## Figures and Tables

**Figure 1 biomedicines-11-02593-f001:**
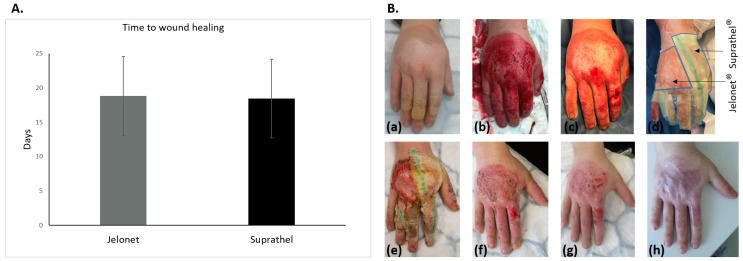
(**A**) Graphical analysis of the data for the time until full wound healing showing no significant difference between Jelonet^®^ and Suprahtel^®^ using a two-sample *t*-test. (**B**) Deep partial-thickness oil burn of the right hand after kitchen accident: (**a**) directly after burn accident, (**b**) after NexoBrid debridement, (**c**) after post-soaking, (**d**) application of the dressings: Jelonet^®^ at the ulnar portion of the back of the hand and Suprahtel^®^ radially and on the fingers (areas marked by black arrows), (**e**) after 8 days, (**f**) after 16 days, (**g**) after 24 days, and (**h**) after 3 months.

**Figure 2 biomedicines-11-02593-f002:**
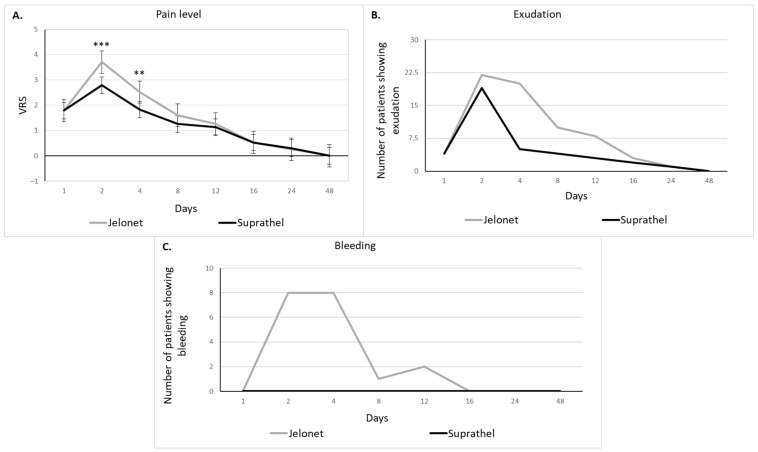
(**A**) Pain level during wound healing (verbal rating scale 0–10) showing significantly less pain after the application of Suprathel*^®^* compared to the use of Jelonet*^®^* on day 2 and day 4 using a two-sample t-test (mean ± SEM; ** indicating *p* < 0.01 and *** indicating *p* < 0.001). (**B**) Exudation during wound healing (analyzed by the visual inspection of an observer who was blinded to the treatment). The area under the curve analysis shows significantly lower exudation after applying Suprathel*^®^* than after applying Jelonet*^®^*. (**C**) Bleeding during wound healing (analyzed by the visual inspection of an observer who was blinded to the treatment). The area under the curve analysis shows significantly lower bleeding rates in the Suprathel^®^ areas with no bleeding being observed.

**Table 1 biomedicines-11-02593-t001:** Patient etiology.

Pat. ID	Gender	Age (y)	TBSA (%)	Burn Cause	Wound Area	Burn Depth	WH S (Days)	WH J (Days)
1	Male	33	0.5	Explosion	Left hand	pt-to-dd	13	13
2	Male	31	1	Flame	Left foot	pt-to-dd	21	28
3	Male	24	5	Scald	Both hands	pt-to-dd	15	18
4	Male	31	23	Scald	Right hand	pt-to-dd	13	13
5	Male	32	2	Scald	Right hand	pt-to-dd	22	22
6	Female	63	8	Oil burn	Right hand	pt-to-dd	13	13
7	Female	48	2	Flame	Both hands	pt-to-dd	24	30
8	Male	21	0.3	Scald	Right hand	pt-to-dd	28	31
9	Male	22	2	Oil burn	Left foot	dd	Skin graft needed	21
10	Male	34	1	Explosion	Both hands	pt-to-dd	11	11
11	Male	35	1.5	Scald	Right hand	pt-to-dd	13	10
12	Male	21	0.8	Flame	Right hand	pt-to-dd	21	21
13	Male	55	2	Flame	Both hands	pt-to-dd	21	21
14	Male	25	1	Scald	Right hand	pt-to-dd	12	12
15	Male	21	2	Explosion	Right hand	pt-to-dd	14	14
16	Male	49	10	Flame	Right hand	pt-to-dd	17	17
17	Female	72	1	Scald	Right hand	pt-to-dd	18	14
18	Female	47	16	Flame	Left hand	pt-to-dd	28	28
19	Trans	54	1.5	Scald	Left hand	pt-to-dd	23	18
20	Male	40	1	Flame	Left hand	pt-to-dd	22	22
21	Male	58	10	Flame	Right hand	pt-to-dd	18	18
22	Male	46	5	Scald	Left hand	pt-to-dd	22	22
23	Male	39	2.5	Scald	Right hand	pt-to-dd	15	15
MEAN		39.17	4.31				18.44	18.81
SD		14.67	5.68				4.78	5.74

Abbreviations: TBSA = total body surface area, WH S = wound healing with Suprathel^®^, WH J = wound healing with Jelonet^®^, pt = partial-thickness burn, and dd = deep dermal burn.

## Data Availability

Data are contained within the article. Additional data presented in this study are available on request from the corresponding author. The data are not publicly available due to privacy issues.

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
