# Peer review of "Comparative Clinical Study of Suprathel® and Jelonet® Wound Dressings in Burn Wound Healing after Enzymatic Debridement"

_biomedicines, 2023, doi:10.3390/biomedicines11102593_

Round 1

Reviewer 1 Report

The authors should elaborate on the novelty of this study. How is their study different from the many similar papers already published in the literature? What is the key element of novelty in the research that is reported?

- The bibliography majorly has references before 2010. The authors should try to cite more relevant and recent literature in the field. such as https://doi.org/10.1080/17425247.2022.2119220.

Did any of the formulations tested reach the desired target in the treatment study? Please explain.

The manuscript does not have a discussion section. This is the most important part for readers understanding. The authors need to add a discussion section.

Author Response

Please see comments in the word document

Reviewer 2 Report

The author's manuscript entitled "Comparison of wound healing, patient comfort, and pain in deep dermal burn wounds treated with Suprathel® and Jelonet® wound dressings after enzymatic debridement with Nexobrid" suggested a comparative clinical study using Suprathel® and Jelonet® wound dressings. While there is relevance to the study in wound healing, still there are concerns/suggestions for the same. 

1. The title is very complex, therefore, needs a simple title such as " Comparative clinical study of Suprathel® and Jelonet® wound dressings in burn wound healing"  2. The figure's quality is very poor, therefore, better quality figures are needed including statistical analysis. Statistical tests should be mentioned under each figure legend. Figure 2 is very poor quality and there are no error bars as well as no statistical information available. 3.  I suggest authors provide relevant and more informative detailed mechanistic comparative data using the preclinical model instead of the clinical trial.  4. There are many limitations of the study, therefore, very few parameters were studied. The discussion is way too lengthy and needs to focus on result specifics.   

Minor editing of English language required

Author Response

Comments in Word document

Reviewer 3 Report

In this report, authors compared the wound healing efficiencies of Suprathel and Jelonet dressings. Authors reported the data from the 23 patients. The study and writing are good however there are some discrepancies in the results and data analysis. The presentation of figures is also need to be improved. Therefore, I would recommend major revision before publication. Here are some comments:

Title of the article is too lengthy. I would recommend to make it concise and specific.

Line 64: Jelonet--à Jelonet®

Line 148-149: Suprathel, Jelonet à Suprathel®, Jelonet®, Make consistent throughout the manuscript.

Figure 1A: Make proper X and Y axis, remove horizontal Grids, the axis title fonts must be same and make them more visible.

Figure 1B, color differences in figure numbering, some black some white. Make them universal.

Figure 1B: Explain the patient aetiology.in this figure, what was the procedure adopted and how both dressings were applied. Mention the portion of wound where the dressing applied using arrows with the name

Line 159: (p = 0.58), This P value is for what? Explain properly. If This P value is in comparison with Suprathel then, it is significantly different.

Line 172-173: if P value is <0.001, then there is no significant difference. How authors concluded that there was significant difference in pain.

Figure 2: the quality of figures is not up to the mark for publication. Add significant difference analysis in the figures (asterisks) for each point wherever the authors find significant difference. Add relevant ANOVA and posthoc test analysis to the caption.

No proper X and Y axis.

Axis titles are invisible

Graph lines are not clear.

While looking at figures 2B and 2C, the bleeding and Exudation events in Jelonet still prominent at day 15th, how authors concluded that both dressings were able to treat the wound in about 18 days.

Mention the active ingredients of both dressings (such as antibiotics, anti-inflammatory or soothing agents) if available.

Author Response

Comments in Word document

Round 2

Reviewer 2 Report

The authors improved the manuscript significantly. Pls see some minor comments for improving Figure 2.

1. Why standard deviations are missing in Fig. 2? 

2. * meaning missing in figure legend.

Author Response

See Word Document

Reviewer 3 Report

I recommend adding sub-numberings to section 2.

Author Response

See Word document
